# OpenReview forum: "Understanding Learning Dynamics of Zeroth-Order Optimization"
_ICLR.cc/2026/Workshop/Sci4DL — Sci4DL 2026_

### Official Review · Reviewer_Vrzr · 2026-02-18

**Fit:** 3
**Significance:** 3
**Confidence:** 2

**Summary:**

Historically, classical optimisation theory has stated that zeroth-order (ZO) algorithms suffer from a dimension-dependent slowdown. According to worst-case results, ZO should be completely useless for massive models like LLMs. Yet, in practice, ZO methods frequently achieve performance comparable with first-order methods. This paper provides a rigorous theoretical explanation for why this empirical success happens using a clever application of Johnson-Lindenstrauss lemma.

**Strengths:**

- The paper is extremely clear. The different steps and concepts are provided precisely when needed.

- While the reformulation of ZO-SGD as an eNTK is straightforward, the application of Johnson-Lindenstrauss lemma to the random projection matrix induced by the perturbations is a clever idea. This allowed to get rid of the dimensionality dependence and bound the discrepancy between first and zeroth order methods on the vocabulary size and the amount of random directions.

**Suggestions:**

- I understand that, given the workshop format, some prioritisation choices had to be made, but I found a rather suboptimal choice presenting the MNIST with LeNet experiment rather than the OPT models. I see the first experiment more as a proof-of-concept rather than an actual contribution, while the second is more significant and validates the theory.

- My second comment comes from someone who, before researching for this paper, was/is very ignorant about zeroth order methods, but I see the current bound based on the number of random projections, still, as a major limitation. Intuitively, for billion-parameter models, the required number of random projections to successfully navigate the search space should theoretically explode. Since the empirical results show it does not (eg P=50 suffices for OPT-1.3B), there appears to be a deeper underlying mechanism at play here that warrants further discussion and explanation

---

### Official Review · Reviewer_Yzjm · 2026-02-25

**Fit:** 3
**Significance:** 3
**Confidence:** 2

**Summary:**

The paper develops a kernel-based theoretical framework to analyze the learning dynamics of Zeroth-Order SGD in function space. The authors derive a projected empirical NTK formulation that characterizes ZO optimization as a random projection of the first-order NTK onto a low-dimensional perturbation subspace. Using this formulation, they show that the discrepancy between ZO and FO learning dynamics is governed by kernel approximation error, which can be bounded using the Johnson–Lindenstrauss lemma. This leads to a key result that ZO optimization fidelity depends primarily on the number of perturbations and output dimension rather than the parameter dimension, providing a theoretical explanation for why ZO methods scale well to large neural networks and LLM fine-tuning. Empirical experiments on LeNet and MNIST demonstrate that increasing perturbations improves alignment between ZO and FO NTK structure, supporting the theoretical analysis.

**Strengths:**

The primary strength of this paper is the introduction of a principled kernel-based perspective for understanding Zeroth-Order optimization learning dynamics. By expressing ZO updates in terms of a projected empirical Neural Tangent Kernel, the paper provides a clear geometric interpretation of ZO optimization as random subspace projection in function space. This perspective is theoretically elegant and helps bridge the gap between optimization-based and kernel-based analyses of neural network training. The derivation of learning dynamics in log-probability space provides a more informative characterization of learning behavior compared to traditional scalar loss analysis.

Another significant contribution is the application of the Johnson–Lindenstrauss lemma to derive approximation guarantees between ZO and FO learning trajectories. The result that kernel approximation fidelity depends on output dimension and number of perturbations rather than parameter dimension provides an important theoretical insight into why ZO methods remain effective in extremely high-dimensional models such as LLMs. This helps resolve a long-standing discrepancy between classical worst-case theoretical scaling and empirical observations of ZO scalability. The theoretical framework is mathematically well motivated and connects ideas from optimization, kernel methods, and random projection theory in a coherent manner.

The paper is also well aligned with the goals of scientific deep learning, as it focuses on understanding the fundamental learning dynamics of optimization methods rather than proposing incremental algorithmic improvements.

**Suggestions:**

One area where the paper can be significantly improved is in the empirical validation of the proposed kernel-based learning dynamics framework in realistic high-dimensional deep learning settings. Currently, the experiments are limited to LeNet on MNIST, which does not reflect the scale, architecture, or optimization characteristics of modern models such as transformers or large language models, where Zeroth-Order optimization is most relevant. In such small-scale settings, the projected kernel approximation may appear accurate even with relatively few perturbations, but this does not adequately test the core theoretical claim that approximation fidelity is independent of parameter dimension. The authors could strengthen their empirical support by evaluating the projected kernel alignment and training dynamics in larger architectures, such as transformer encoders or decoder-only language models, and by analyzing how kernel fidelity and convergence behavior scale with perturbation count, output dimension, and model size.

Another limitation is that the paper focuses primarily on kernel approximation fidelity as a theoretical proxy for learning dynamics but does not directly demonstrate how this approximation impacts practical optimization performance. It remains unclear whether improved alignment between projected and full kernels translates into faster convergence, improved training stability, or better final performance. To address this, the authors could empirically correlate kernel alignment metrics with training loss reduction, convergence rate, or generalization performance across different perturbation budgets. Additionally, since neural network gradients often lie in a structured low-dimensional subspace rather than being uniformly distributed, it would be valuable to investigate whether the effectiveness of random perturbations depends on the intrinsic gradient subspace structure. Analyzing how projected perturbations preserve dominant gradient directions or principal components would provide deeper insight into when and why Zeroth-Order optimization remains effective in practice.

---

### Official Review · Reviewer_CSwQ · 2026-02-27

**Fit:** 3
**Significance:** 2
**Confidence:** 2

**Summary:**

This work offers a function space perspective on zeroth-order optimization through the empirical Neural Tangent Kernel. The paper derives one-step learning dynamics of ZO-SGD, with the key observation that the ZO gradient update corresponds to a projection of the FO gradient onto a random subspace spanned by the perturbation vectors, giving rise to a ZO-eNTK. Using the Johnson-Lindenstrauss Lemma, the authors bound the discrepancy between ZO and FO kernels and show that the approximation error depends on the output vocabulary size V rather than the parameter dimension d. This parameter-dimension-free property offers an explanation for why ZO methods remain effective for very large models. The analysis covers both single and multi-perturbation regimes and includes some empirical validation. This is a solid theoretical paper whose main practical takeaway is a principled criterion for how many perturbations are needed to achieve FO-comparable dynamics, though it feels cut short by a missing conclusion section.

**Strengths:**

- The identification of the ZO-eNTK as a projection of the FO-eNTK onto the subspace spanned by perturbation vectors is an insightful and valuable contribution. It provides a clean geometric picture of what ZO optimization is actually doing in function space, which, to my knowledge, has not been articulated this way before.
- The (parameter)dimension-free approximation result follows naturally from this geometric view and is a notable finding, as the dependence on V rather than d offers a clearer explanation for ZO applicability/validity than earlier works.
- The empirical validation, including results on LLMs of varying scale, provides good support for the theoretical findings.

**Suggestions:**

- In the discrepancy bound (Appendix D.1.2, Eq. 13-15), the final rate includes a Jacobian/gradient-magnitude prefactor that is not analyzed. Outside the NTK regime, this term can grow with model size and effectively reintroduce parameter-dimension dependence, weakening the main dimension-free claim. It would strengthen the paper to either explicitly bound this term, state the conditions under which it remains controlled, or provide measurements showing it does not grow with model size in the fine-tuning settings considered. Did the authors account for how this prefactor behaves as the model scales (assumptions, a separate bound, or measurements), and how sensitive the conclusions are if it is not controlled?
- The analysis is explicitly one-step, and the paper is upfront about this. However, a brief comment on multi-step closeness would be a useful addition. In the worst case, per-step discrepancies accumulate over training, but under a lazy training or fine-tuning regime where the kernel changes little between steps, errors may remain controlled. Even an informal argument or an empirical measurement of how much the ZO and FO trajectories diverge over a full training run would strengthen the paper's practical claims.
- The paper seems to be missing a proper conclusion section.
- (minor): Figure 2 caption would benefit from a more detailed description.

---

### Meta-Review · Area_Chair_eyvA · 2026-03-01

**Recommendation:** Accept

**Metareview:**

The work focuses on learning behavior of zeroth order optimization and makes interesting connections to NTK. My recommendation is accept.

---

### Decision · Program_Chairs · 2026-03-02

Accept